# The Sympathetic Nervous System Regulates Sodium Glucose Co-Transporter 1 Expression in the Kidney

**DOI:** 10.3390/biomedicines11030819

**Published:** 2023-03-07

**Authors:** Jennifer Matthews, Moira Hibbs, Lakshini Herat, Markus Schlaich, Vance Matthews

**Affiliations:** 1Dobney Hypertension Centre, School of Biomedical Science—Royal Perth Hospital Unit, Royal Perth Hospital Research Foundation, University of Western Australia, Crawley, WA 6009, Australia; 2Research Centre, Royal Perth Hospital, Perth, WA 6000, Australia; 3Dobney Hypertension Centre, Medical School—Royal Perth Hospital Unit, Royal Perth Hospital Research Foundation, University of Western Australia, Crawley, WA 6009, Australia; 4Department of Cardiology and Department of Nephrology, Royal Perth Hospital, Perth, WA 6000, Australia

**Keywords:** SGLT1, hypertension, diabetes, mice, cells, kidney

## Abstract

Hyperactivation of the sympathetic nervous system (SNS) has been demonstrated in various conditions including obesity, hypertension and type 2 diabetes. Elevated levels of the major neurotransmitter of the SNS, norepinephrine (NE), is a cardinal feature of these conditions. Increased levels of the sodium glucose cotransporter 1 (SGLT1) protein have been shown to occur in the parotid and submandibular glands of hypertensive rodents compared to normotensive controls. However, there was a need to examine SGLT1 expression in other tissues, such as the kidneys. Whether NE may directly affect SGLT1 protein expression has not yet been investigated, although such a link has been shown for sodium glucose cotransporter 2 (SGLT2). Hence, we aimed to determine (i) whether our murine model of neurogenic hypertension displays elevated renal SGLT1 expression and (ii) whether NE may directly promote elevations of SGLT1 in human proximal tubule (HK2) cells. We did indeed demonstrate that in vivo, in our mouse model of neurogenic hypertension, hyperactivation of the SNS promotes SGLT1 expression in the kidneys. In subsequent in vitro experiments in HK2 cells, we found that NE increased SGLT1 protein expression and translocation as assessed by both specific immunohistochemistry and/or a specific SGLT1 ELISA. Additionally, NE promoted a significant elevation in interleukin-6 (IL-6) levels which resulted in the promotion of SGLT1 expression and proliferation in HK2 cells. Our findings suggest that the SNS upregulates SGLT1 protein expression levels with potential adverse consequences for cardiometabolic control. SGLT1 inhibition may therefore provide a useful therapeutic target in conditions characterized by increased SNS activity, such as chronic kidney disease.

## 1. Introduction

The sympathetic nervous system (SNS) is responsible for the ‘fight or flight’ response. Upon its activation, blood flow from the skin and the gastrointestinal tract is redirected to the brain, heart and lungs [1]. At rest, the SNS, composed of preganglionic neurons originating from the spinal cord, maintains homeostatic blood pressure, body temperature and blood glucose levels through a hormonal cascade. These preganglionic neurons secrete acetylcholine to activate the sympathetic postganglionic neurons or specialised cells in the adrenal gland. Once activated, the major neurotransmitters of the SNS, norepinephrine (NE) or epinephrine, are secreted to target specific organs or tissues, which express either alpha or beta-adrenergic receptors [2]. This enables the effective binding of the catecholamines to the specific organ or tissue sites. Overactivity of the SNS has been associated with adverse metabolic consequences in various common conditions such as obesity, hypertension and diabetes mellitus [3,4,5].

Recent studies have demonstrated the critical role of Sodium Glucose Co-transporters (SGLT’s) in metabolic, cardiac and renal disease [6]. The exact mechanisms underlying the benefits of Sodium Glucose Co-transporter 2 inhibitors (SGLT2i) in providing cardiovascular and renal protection remain to be fully elucidated [7]. We have previously shown a link between Sodium Glucose Co-transporter 2 (SGLT2) expression and its modulation by the SNS [3,4]. The potential interaction between Sodium Glucose Co-transporter 1 (SGLT1) and SNS has not yet been explored.

Although the function of SGLT1 is identical to that of SGLT2, it is more widely expressed throughout the body including the small intestine, eye, kidney, liver, pancreas and heart [6].

Currently, there is limited information on the regulation of SGLT1, particularly in the context of a potential interplay with the SNS. We, therefore, aimed to determine whether the SNS regulates SGLT1 expression. This was achieved by (i) measuring SGLT1 expression in the kidneys of our neurogenically hypertensive mice and (ii) treating human proximal tubule (HK2) cells with NE and subsequently assessing SGLT1 expression and associated mechanisms. We hypothesised that SNS activation may upregulate SGLT1 expression. This study may ultimately highlight that SGLT1 inhibition may be beneficial to treat diseases associated with SNS activation such as chronic kidney disease.

## 2. Materials and Methods

### 2.1. Animals

Renal tissue was collected from 15-week-old male and female blood pressure normal (BPN/3J) mice or blood pressure high (BPH/2J, also known as Schlager) mice [3,8]. Our BPH/2J Schlager mice with neurogenic hypertension [9,10] are a highly relevant mouse model since they mimic human disease with increased sympathetic activity [9], elevated heart rate and heightened blood pressure [8,11] driven by neurogenic mechanisms [12]. All dissections were carried out at the Royal Perth Hospital (RPH) animal holding facility in accordance with the guidelines of the RPH Animal Ethics Committee (R537/17-20, approval date 17 August 2017). Kidney tissue was fixed in paraformaldehyde and subsequently embedded in paraffin wax as reported by Herat et al. [3].

### 2.2. Immunohistochemistry of Tyrosine Hydroxylase (TH) in Renal Tissue

We performed immunohistochemistry in the kidneys of our normotensive BPN/3J mice and hypertensive BPH/2J mice. Kidney tissues from both our BPN/3J and BPH/2J mice were sectioned at 5 μm onto positively charged microscope slides and de-waxed in xylene and rehydrated in ethanol. Antigen retrieval was performed on the slides by heating in EDTA buffer (pH 8.5; Sigma-Aldrich, Sydney, NSW, Australia). Slides were treated with 3% hydrogen peroxide and then blocked in 5% FCS in PBS/0.1% Tween-20. Tyrosine hydroxylase was detected with rabbit anti-tyrosine hydroxylase antibody (AB152; Merck Millipore, Melbourne, VIC, Australia). Antibody binding was detected with anti-rabbit (1:100, Santa Cruz Biotechnology, Sydney, NSW, Australia) secondary antibodies conjugated to HRP, followed by treatment with diaminobenzidine (DAB, Ventana, AZ, USA). Tissues were counterstained with haematoxylin before being dehydrated in ethanol and cleared in xylene and mounted with DPX (Sigma-Aldrich, Sydney, NSW, Australia). Photomicrographs were taken of stained kidneys from mice using a Nikon Eclipse Ti Microscope (Nikon Instruments Inc, Tokyo, Japan). Tyrosine hydroxylase positive nerves were counted in random fields of view.

### 2.3. SGLT1 Immunohistochemistry of Renal Tissues

We performed immunohistochemistry in the kidneys of normotensive BPN/3J and hypertensive BPH/2J mice. Briefly, kidney tissue was fixed in 10% buffered formalin for 24 h, followed by wax embedding. Paraffin sections (5 μm) were collected and mounted on slides. For antigen retrieval, slides were heated for 2.5 min in a pre-heated 1× EDTA buffer (pH 8.5; Sigma-Aldrich, Sydney, NSW, Australia). After washing twice in PBS/0.1% Tween for 5 min, tissue sections were outlined with a paraffin pen. Sections were blocked with 3% H_2_O_2_ for 10 min, washed twice with PBS/0.1% Tween for 5 min and blocked with 5% FCS in PBS/0.1% Tween for 1 h in a humidified chamber. Sections were then incubated overnight at 4 °C in a humidified chamber with both rabbit anti-SGLT1 antibody (1:180; Abcam, Sydney, NSW, Australia) and rabbit anti-SGLT1 antibody (1:100; Novus, Sydney, NSW, Australia) in 5% FCS/PBS/0.1% Tween. Following overnight incubation, sections were washed three times with PBS/0.1% Tween for 5 min and incubated with anti-rabbit (1:100, Santa Cruz Biotechnology, Sydney, NSW, Australia) secondary antibodies conjugated with HRP in PBS/0.1% Tween for 1 h. This was followed by incubation with diaminobenzidine (DAB, Ventana, AZ, USA). Slides were counterstained with hematoxylin, dehydrated, and mounted with DPX (Sigma-Aldrich, Sydney, NSW, Australia). Photomicrographs were taken of stained kidneys from mice using a Nikon Eclipse Ti Microscope (Nikon Instruments Inc, Tokyo, Japan). The intensity of proximal tubule staining in each field of view was rated against a scale of 0–3 (0 = no staining; 1 = low staining; 2 = intermediate staining; 3 = high intensity of staining).

### 2.4. Human Kidney 2 Cell Culture

The human renal proximal tubule cell line, HK2, was generously provided by Dr Melinda Coughlan and Prof Karin Jandeleit-Dahm (Baker IDI Heart and Diabetes Institute, Melbourne, VIC, Australia). Cells were cultured in high-glucose Dulbecco’s Modified Eagle Medium (HG-DMEM) supplemented with L-glutamine (1%), streptomycin/penicillin (2%) and fetal calf serum (FCS) (10%) (Thermo Fisher, Melbourne, VIC, Australia).

Unless stated otherwise, cells were trypsinized, plated in Corning Cell Bind 6 well plates (Corning, Glendale, AZ, USA) and allowed to grow to 70% confluency before being treated with NE (Sigma-Aldrich, Sydney, NSW, Australia). Norepinephrine was diluted in Baxter water and added to the wells in accordance with previous studies [4]. Control wells were treated with Baxter water. Norepinephrine was protected from light during its preparation and experiments.

### 2.5. Immunocytochemistry

SGLT1 was detected in HK2 cells using the immunocytochemistry technique. Using a 6 well plate, the cells were fixed in methanol/acetone (1:1) and then endogenous peroxidases were blocked with 3% H_2_O_2_ followed by blocking with 10% FCS/Tx/PBS. The primary antibody (1:500, Rabbit anti-SGLT1 antibody, Novus, Centennial, CO, USA.) diluted in Triton X/PBS was added to each well and incubated at 4 °C overnight. The following day a secondary conjugated antibody (1:100 goat anti-Rabbit IgG Peroxidase, Thermofisher, Melbourne, VIC, Australia) was added to each well and then stained with DAB prior to visualisation with a high-powered microscope.

### 2.6. Sodium Glucose Co-Transporter-1 Translocation in HK2 Cells Treated with NE or Hyper Interleukin 6 (H-IL-6)

HK2 cells were seeded into six-well Cell Bind plates and treated with NE (10 μΜ) for 48 h. Cells were permeablized and separated into cytoplasmic, cytoskeletal and membrane protein fractions using the Subcellular Protein Fractionation Kit for Cultured Cells (Cat#: 78840, Thermo Fisher Scientific, Carlsbad, CA, USA). Extracts were analysed using human SGLT1 ELISA (Cloud-Clone, Wuhan, China). Alternatively, HK2 cells were treated with Hyper IL-6 (10 ng/mL–1000 ng/mL; a kind gift from Prof. Stefan Rose-John, University of Kiel, Germany) acutely (15 min) or chronically (24 h).

### 2.7. Interleukin-6 (IL-6) and Tumor Necrosis Factor Alpha (TNF-α) Secretion in Human Kidney 2 Cells Treated with NE

Cells were seeded into six-well Cell Bind plates and treated with NE. The media from each well was sampled at 24 h. Samples were centrifuged at 2100 rpm for 10 min and the cell-free culture media supernatants were stored at −80 °C until analysis was conducted using a Quantikine human IL-6 ELISA kit (Cat#: D6050, R&D System, Minneapolis, MN, USA) or human TNF-α ELISA kit (Elisakit.com, Product 0005, Melbourne, VIC, Australia).

### 2.8. Western Blotting

Phosphorylated STAT-3 was detected using anti-phospho STAT-3 antibody (Tyr-705; Cat#: 9131, Cell Signaling Technology, Danvers, MA, USA) followed by goat anti-rabbit 800 antibody (Cat#: 926-32211, LiCor, Lincoln, NE, USA). Beta-actin was detected using mouse anti-beta-actin antibody (Cat#: ab6276, Abcam, Sydney, NSW, Australia) followed by anti-mouse 680 antibody (Cat#: 926-68020, LiCor, Lincoln, NE, USA). Cyclin Dependent Kinase 4 (CDK4) was detected using mouse anti-CDK4 antibody (Cat#: AHZ0202, Invitrogen, Sydney, NSW, Australia) followed by anti-mouse 680 antibody (Cat#: 926-68020, LiCor, Lincoln, NE, USA). Imaging was performed using the Odyssey detection apparatus (Licor, Lincoln, NE, USA).

### 2.9. Enzyme-Linked Immunosorbent Assays

Frozen Kidneys were homogenized and analysed for NE content using the mouse norepinephrine NE ELISA kit (CSB-E07870m; Cusabio, Wuhan, China) according to the manufacturer’s instructions.

### 2.10. Statistical Analysis

All quantitative data is presented as mean ± SEM. A significance level of *p* value less than 0.05 was considered significant. Significance was determined for all data using Students *t*-tests. Graphs were generated using GraphPad Prism 9 (GraphPad Software, San Diego, CA, USA).

## 3. Results

### 3.1. Tyrosine Hydroxylase (TH) Is Increased in the Kidneys of BPH/2J Neurogenically Hypertensive Mice in Comparison to BPN/3J Normotensive Mice

In our current study, we used a neurogenic hypertensive mouse model, known as the Schlager (BPH/2J) mouse which has a heightened sympathetic tone. In order to elucidate that the hypertensive model has sympathetic hyperactivity compared to the normotensive BPN/3J model, we investigated the renal TH protein levels in both strains and found that there is a significant increase in TH positive punctate staining in the neurogenically hypertensive BPH/2J mice (Figure 1).

### 3.2. Norepinephrine Levels Are Significantly Elevated in the Kidneys of BPH/2J Neurogenically Hypertensive Mice in Comparison to BPN/3J Normotensive Mice

We have shown that the marker of sympathetic hyperactivity, norepinephrine, is significantly elevated in the BPH/2J mice when compared to our BPN/3J normotensive mice (Figure 2).

### 3.3. SGLT1 Protein Levels Are Increased in BPH/2J Neurogenically Hypertensive Mice in Comparison to BPN/3J Normotensive Mice

The protein SGLT1 is currently studied as a therapeutic target for the treatment of diabetes [13]. We observed that neurogenically hypertensive BPH/2J mice displayed significantly upregulated SGLT1 protein expression in the kidney compared to the normotensive BPN/3J mice (Figure 3).

### 3.4. Norepinephrine Promotes Increased Expression and Translocation of SGLT1

Norepinephrine (NE) is the major neurotransmitter of the SNS. Using our previously optimised concentration of NE in HK2 cells, we have previously demonstrated that NE increased SGLT2 expression (4). In our current study, we demonstrated for the first time that NE reproducibly resulted in increased SGLT1 expression in HK2 cells (Figure 4A). We then wanted to determine whether NE induced SGLT1 protein translocates to the cell membrane where it may be functional. Using subcellular fractionation to isolate the membrane fraction, we indeed observed that NE promoted SGLT1 to translocate to the cell membrane (Figure 4B).

To further supplement the above findings in Figure 4, we also conducted SGLT1 immunocytochemistry. In support of the quantitative SGLT1 ELISA data, we demonstrated that NE consistently elevates SGLT1 protein which is evident in the cytoplasm of the HK2 cells (Figure 5).

### 3.5. Norepinephrine Treatment Increases IL-6 Secretion from HK2 Cells

As NE increased SGLT1, we then wanted to ascertain upstream mediators. It is known that TNF-α and IL-6 may regulate SGLT1 (either in concert with other cytokines or individually; [14,15]). Therefore, we determined the secreted levels of TNF-α and IL-6 in our in vitro experiments. Interestingly, TNF-α secretion was not different between vehicle and NE treated cells. However, NE profoundly upregulated IL-6 release (Figure 6). Therefore, we wanted to further understand whether IL-6 may directly promote upregulation of SGLT1.

### 3.6. Hyper IL-6 Is Bioactive in HK2 Cells and Promotes Increased SGLT1 Expression

As conditions involving hyperactivation of the SNS such as type 2 diabetes (T2D) and hypertension display increased levels of IL-6 and its soluble IL-6 receptor (sIL-6R) [14,16,17], we utilised the IL-6 and sIL-R fusion protein, hyper IL-6. All cells express GP130 which is necessary for IL-6/sIL-6R signalling. Concentrations of 100 ng/mL and above were bioactive in HK2 cells as evidenced by increased STAT-3 phosphorylation (Figure 7).

We were able to also demonstrate that Hyper IL-6 elevates SGLT1 expression in HK2 cells (Figure 8).

### 3.7. Hyper IL-6 Promotes Proliferation of HK2 Cells and Increases Cyclin Dependent Kinase 4 (CDK4) Levels

In addition to Hyper IL-6 upregulating SGLT1, we consistently observed that the IL-6/sIL-6R fusion protein promoted proliferation of HK2 cells (Figure 9). This is a highly novel finding.

As further evidence that hyper IL-6 promotes proliferation of HK-2 cells, we looked at expression of the 34 kD protein CDK4. We found that hyper IL-6 significantly upregulated CDK4 protein levels (Figure 10). This result, together with the proliferation data (Figure 9) provides strong evidence that hyper IL-6 may promote proliferation of proximal tubule cells during chronic kidney disease.

## 4. Discussion

Hypertension and DM are chronic conditions that cumulatively affect more than two billion people globally every year [18,19]. Blood glucose levels and blood pressure are regulated by the SNS via the regulation of liver, pancreas and kidney function and peripheral vasculature tone [20]. The high incidence, mortality rates and conditions associated with hypertension and diabetes have highlighted that further research regarding the management and treatment of these diseases is urgently required.

The interplay between obesity and hypertension is evident since obesity dysregulates the SNS via various pathways including an increase in renal SNS activity [19]. This in turn affects the renin-angiotensin-aldosterone system and overall increases sodium retention and promotes hypertension [21]. Furthermore, there is a correlation between diabetes and hypertension since it has been established that more than 50% of people with diabetes are also hypertensive [22].

Various approaches are available to treat neurogenic hypertension. Moxonidine is a centrally acting selective imidazoline receptor agonist (SIRA) that reduces peripheral sympathetic activity and, therefore, decreases peripheral vascular resistance [23]. Surgically, renal denervation utilises radiofrequency ablation to ablate renal nerves [24], leading to a suppression of SNS activity and thereby decreasing blood pressure and renal glucose reabsorption [24]. The decrease in renal glucose reabsorption may be due to the fact that we and others have shown that (i) the major neurotransmitter of the SNS, NE, is critical for promoting expression of SGLT’s which is relevant for glucose reabsorption. Of the two main family members, Sodium Glucose Co-transporter 2 (SGLT2) has been widely studied for decades and has been found to be regulated by the SNS [6], whereas Sodium Glucose Co-transporter 1 (SGLT1) is less well investigated. Rafiq et al. (2015) demonstrated that NE, which is the major neurotransmitter of the SNS, may increase *Sglt2 mRNA* [25]. However, SGLT2 protein was not assessed. Our group [4] displayed for the first time that SGLT2 protein expression can be positively regulated by the SNS in human proximal tubule (HK2) cells [4]. In these experiments, HK2 cells were treated with various concentrations of NE to mimic the activation of the SNS. The findings demonstrated a dose-response regarding NE treatment and SGLT2 expression. With these findings in mind, our current study aimed to determine whether the activity of the SNS also influenced SGLT1 expression. This hypothesis is further supported by our current in vivo findings which suggest that a hyperactive SNS may also be associated with increased SGLT1 protein expression.

In our current study, we show for the first time in our BPN/3J and BPH/2J mice, that the neurogenically hypertensive BPH/2J mice have significantly increased TH and NE levels in the kidney, and this promoted renal SGLT1 expression. Together, the elevated NE and TH levels in the BPH/2J mice support the notion that this strain displays SNS hyperactivation compared to the BPN/3J strain. The increased TH and NE levels in the BPH/2J mice correlate with our previously published blood pressure results where we highlight that BPH/2J mice display neurogenic hypertension [3]. The increased renal SGLT1 expression in the BPH/2J mice highlights that SGLT1 may be a therapeutic target. Furthermore, previous studies utilising the spontaneous hypertensive rats (SHR) and the Wistar Kyoto rats (WKY) showed that the sympathetic nerve activity and SGLT1 levels are elevated in the salivary glands of the hypertensive rodents [26]. In our study, we performed immunohistochemistry in the kidneys of the normotensive BPN/3J mice and neurogenically hypertensive BPH/2J mice and found that SGLT1 was specifically elevated in the BPH/2J kidneys (Figure 3). We then followed this result up with novel in vitro studies which aimed to determine if increased SNS activity may directly affect SGLT1 protein levels.

Norepinephrine is one of the primary markers of SNS activation. We investigated whether NE increased SGLT1 expression in the HK2 cells which has already been shown to express SGLT1 [27]. In our current study, we highlight a statistically significant increase in SGLT1 protein expression after NE treatment when using both specific immunohistochemistry and a specific SGLT1 ELISA (Figure 4 and Figure 5). This provides strong evidence that the SNS positively regulates the SGLT1 protein which may be pathogenic if increased too greatly as it may promote hypertension and glucose reabsorption.

Two of the prominent pro-inflammatory cytokines, TNF-α and IL-6 are elevated in conditions resulting from heightened SNS activation, such as T2D [14,15,16] and hypertension [28,29]. While TNF-α has been shown to decrease the SGLT1 expression in the brush border membrane of the intestine [30,31,32], IL-6 has been shown to increase the SGLT1 expression [33]. With TNF-α and IL-6 already being known to be involved in the regulation of SGLT1, we wanted to ascertain the effect that NE had on these upstream mediators. Although TNF-α was not differentially regulated by NE, IL-6 was significantly upregulated (Figure 6), therefore prompting further experimentation involving IL-6. As both IL-6 [14,16,17] and soluble IL-6 receptor (sIL-6R) [17] are increased in T2D, we utilised the hyper IL-6 fusion protein (IL-6 and sIL-6R), which resembles the pathogenic IL-6 mediated effects in T2D. In our study, we uncover a highly novel finding that demonstrates that in the HK2 cell line, Hyper IL-6 not only promoted elevated expression of SGLT1 but also promoted proliferation of the cells (Figure 8, Figure 9 and Figure 10). Our findings are biologically relevant as the proximal tubule is particularly targeted during acute and chronic kidney injuries [34].

A marker of cell proliferation is cyclin D-dependent kinase 4 (CDK-4) which helps to drive the progression of cells into the DNA synthesis phase of the cell division cycle. We found that hyper IL-6 significantly upregulated CDK4 levels in HK2 cells (Figure 10). Interestingly, CDK4 mediated proliferation has been associated with pathogenicity. It is likely that blocking CDK4 will prevent damaged proximal tubule cells from proliferating during kidney injury. Injured proximal tubule cells promote tubulointerstitial fibrosis through production of proinflammatory and profibrotic cytokines [34]. In conclusion, blocking cell cycle progression by inhibiting CDK4 may protect against chronic kidney disease by preventing damaged proximal tubule cells from proliferating by reducing tubular injury, fibrosis and senescence [34].

## 5. Conclusions

In this novel study, we demonstrated that the SNS promotes SGLT1 expression in the kidneys in vivo. We also found that the major neurotransmitter of the SNS, NE, increased SGLT1 protein expression and translocation to the cell surface in vitro. Additionally, NE promoted a significant elevation in interleukin-6 (IL-6) levels which resulted in the promotion of SGLT1 expression and proliferation in HK2 cells. Our findings suggest that the SNS upregulates SGLT1 protein expression levels with potential adverse consequences for cardiometabolic control.

There is a current lack of studies examining the transcription and translation of SGLT1. Therefore, it would be highly interesting to examine the transcriptional and translational regulatory mechanisms underlying SNS induced increases in SGLT1. Future studies should look at utilization of Sodium Glucose Co-transporter 1 inhibitors (SGLT1i) such as KGA2727 in diseases where the SNS is dysregulated. SGLT1 inhibition may, therefore, provide a useful therapeutic target in conditions characterized by increased SNS activity, such as chronic kidney disease.

## Figures and Tables

**Figure 1 biomedicines-11-00819-f001:**
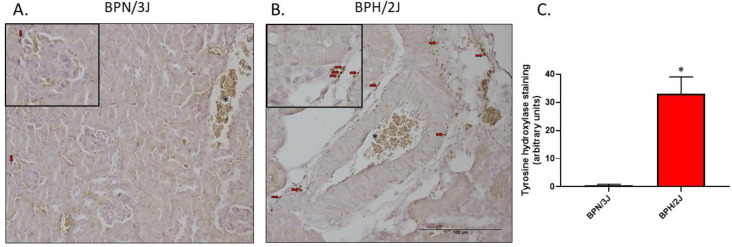
Tyrosine Hydroxylase is significantly increased in the kidneys of BPH/2J neurogenically hypertensive mice compared to BPN/3J normotensive mice. Representative Tyrosine Hydroxylase immunohistochemistry images of renal tissue of (**A**) normotensive mice or (**B**) neurogenically hypertensive mice, * in (**A**) and (**B**) indicates erythrocytes. (**C**) Quantitation of TH staining in kidneys. * *p* = 0.0017; data represented as mean ± SEM; *n* = 4/group. Red arrows = TH staining; Bar = 100 μm (as indicated by image (**B**)).

**Figure 2 biomedicines-11-00819-f002:**
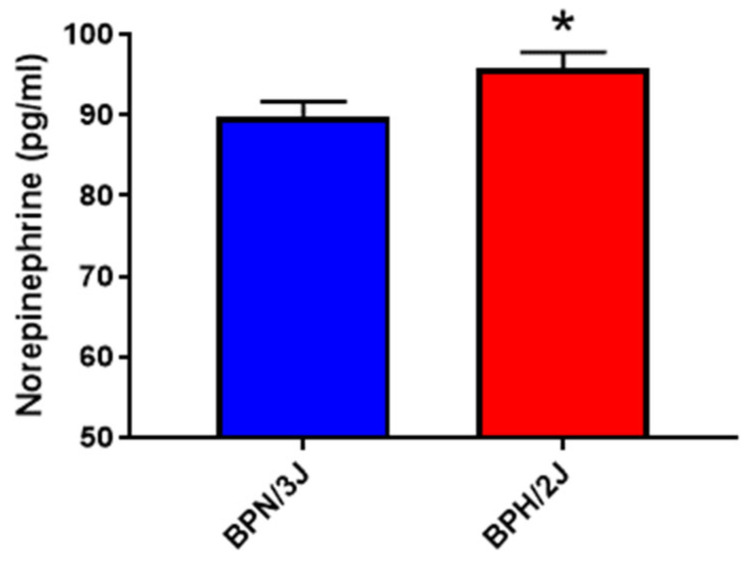
Norepinephrine levels are significantly increased in the kidneys of BPH/2J neurogenically hypertensive mice compared to BPN/3J normotensive mice. Quantitation of NE levels in kidneys using a specific mouse NE ELISA. * *p* = 0.04; data represented as mean ± SEM; *n* = 13–14/group.

**Figure 3 biomedicines-11-00819-f003:**
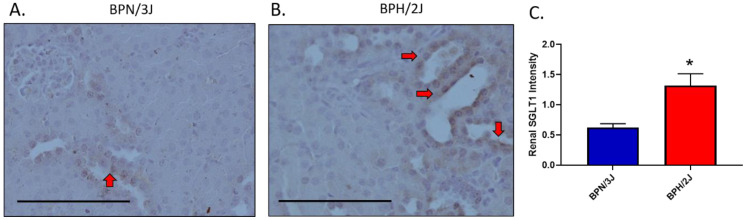
Hyperactivation of the SNS promotes upregulation of SGLT1 protein levels in the kidneys of neurogenically hypertensive BPH/2J mice. Representative SGLT1 immunohistochemistry images of renal tissue of (**A**) normotensive mice or (**B**) neurogenically hypertensive mice (**C**) Quantitation of SGLT1 intensity. * *p* = 0.037; data represented as mean ± SEM; *n* = 3–5/group. Red arrows = luminal SGLT1 expression; Bar = 100 μm; SGLT1, Sodium glucose cotransporter 1.

**Figure 4 biomedicines-11-00819-f004:**
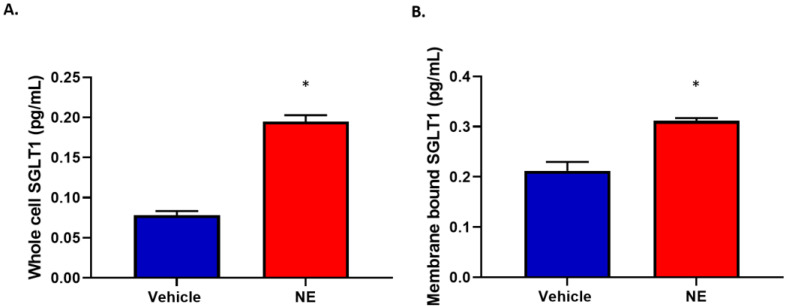
SGLT1 protein levels are elevated and translocate to the cell membrane following NE (10 μM) treatment of HK2 cells. (**A**) whole cell lysates; data represented as mean ± SEM; n = 5/group; * *p* = 0.0000019 (**B**) membrane bound SGLT1; data represented as mean ± SEM; *n* = 3/group; * *p* = 0.006; SGLT1, Sodium glucose cotransporter 1; NE, Norepinephrine.

**Figure 5 biomedicines-11-00819-f005:**
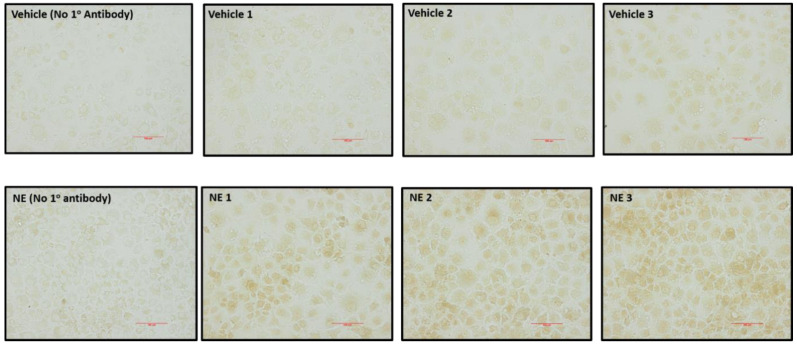
Immunocytochemical detection of NE induced SGLT1 expression in HK2 cells. Each image indicates an individual well of HK2 cells. SGLT1 staining is brown in colour. Size bar = 100 μm.

**Figure 6 biomedicines-11-00819-f006:**
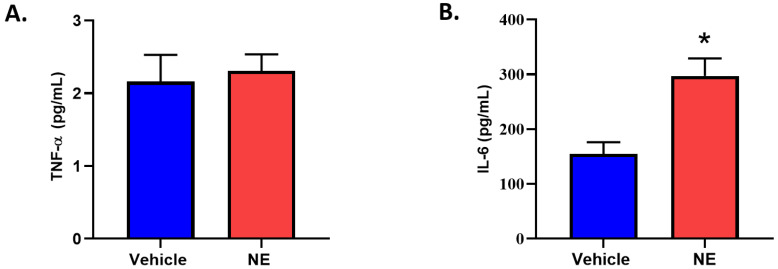
NE (10 μm) treatment of HK2 cells promoted a significant elevation in IL-6 secretion but did not affect TNF-α secretion. (**A**) TNF-α levels in the supernatant of HK2 cells treated with Vehicle or NE for 24 h; data represented as mean ± SEM; *n* = 5/group. (**B**) IL-6 levels in the supernatant of HK2 cells treated with Vehicle or NE for 24 h; data represented as mean ± SEM; *n* = 5–6/group; * *p* = 0.0065.

**Figure 7 biomedicines-11-00819-f007:**
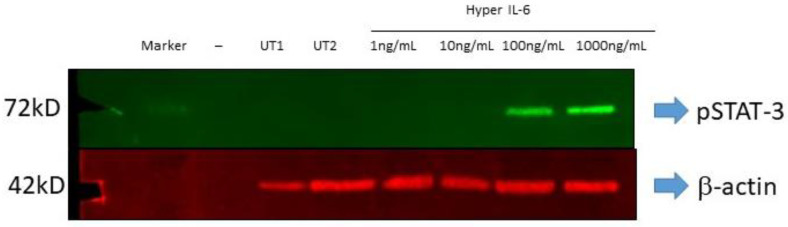
Phosphorylated STAT-3 levels in HK2 cells after being treated with Hyper IL-6.

**Figure 8 biomedicines-11-00819-f008:**
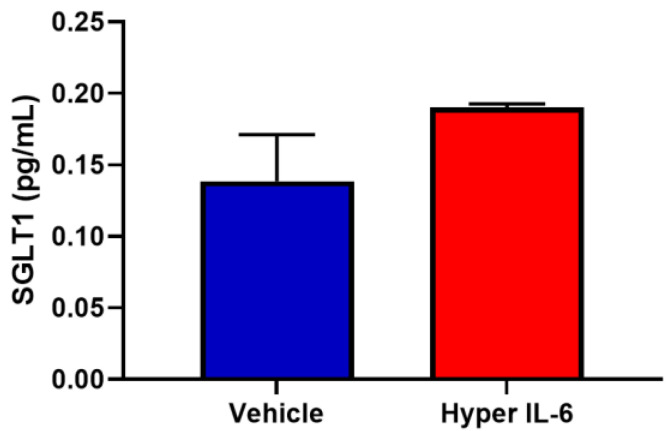
Hyper IL-6 promotes SGLT1 expression in HK2 cells. Cells were treated with vehicle or Hyper IL-6 for 24 h; data represented as mean ± SEM; *n* = 3/group.

**Figure 9 biomedicines-11-00819-f009:**
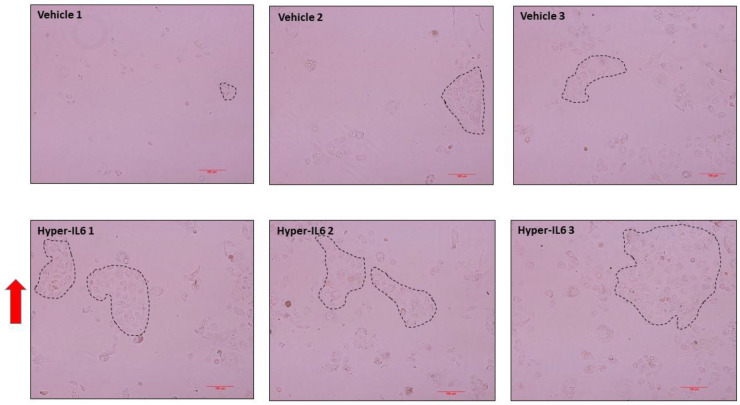
Hyper IL-6 promotes proliferation of HK2 cells after 24 h of treatment; Each image indicates a representation from an individual well. Dashed lines indicate areas of cellular foci; *n* = 3/group. Size bar = 100 μm. Red arrow indicates increased proliferation in Hyper-IL6 treated cells.

**Figure 10 biomedicines-11-00819-f010:**
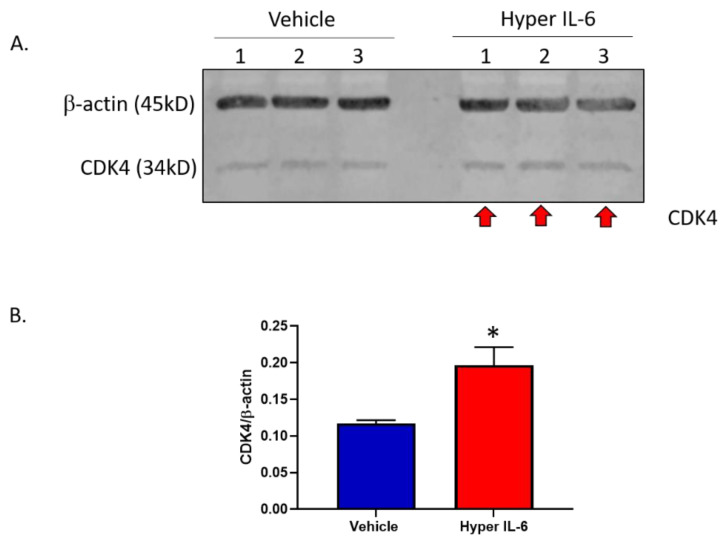
Hyper IL-6 promotes increased Cyclin dependent kinase 4 (CDK4) protein expression in HK2 cells. (**A**) CDK4 western blot and (**B**) Quantitation of CDK-4 protein relative to the housekeeper protein β-actin. Cells were treated for 48 h.* *p* = 0.034. Data represented as mean ± SEM; *n* = 3/group. Red arrows indicate increased CDK4 expression.

## Data Availability

All requests for data should be made to the corresponding author.

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
