# Peer review of "The Sympathetic Nervous System Regulates Sodium Glucose Co-Transporter 1 Expression in the Kidney"

_biomedicines, 2023, doi:10.3390/biomedicines11030819_

Round 1

Reviewer 1 Report

This study demonstrated that the SNS promotes SGLT1 expression in the kidneys in vivo. The findings of this study suggest that the SNS upregulates SGLT1 protein expression levels with potential adverse consequences for cardiometabolic control. Therefore, I recommend the publication of this manuscript after the following minor revisions.

1.       All abbreviations should be defined when they first appear in the text.

2. More about future remarks of this study should be added in the conclusion.

Reviewer 2 Report

Dear Editor, I’ve read with great interest the draft “The Sympathetic Nervous System regulates Sodium Glucose Co-transporter 1 expression in the kidney” by Jennifer Matthews et al.

However, some issues need to be raised:

Introduction:

-          The authors write “Overactivity of the SNS has been associated with… diabetes mellitus”, this sentence should be associated with a reference.

-          The authors write “The exact mechanisms underlying the benefits of Sodium Glucose Co-transporter 2 inhibitors (SGLT2i) … remain to be fully elucidated”, this sentence should be associated with a reference. I would suggest the recent “An Overview of the Cardiorenal Protective Mechanisms of SGLT2 Inhibitors.”  Published 2022 Mar 26. doi:10.3390/ijms23073651

-          The authors write “We have previously shown… modulation by the SNS”, this sentence should be associated with a reference.

Results:

-          The authors write: “Together, the elevated NE and TH levels in the BPH/2J mice supports… BPN/3J strain.” In the opinion of this reviewer, this sentence here is redundant, I would move into Discussion section.

-           The authors write: “suggesting that a hyperactive SNS may be associated with increased SGLT1 protein expression”.  In the opinion of this reviewer, this sentence here is redundant, I would move into Discussion section.

-          The authors write: “This is a highly novel finding.” I would move into Discussion section.
